# Interactions between *Mycoplasma mycoides* subsp. *mycoides* and bovine macrophages under physiological conditions

**Philippe Totté\*, Tiffany Bonnefois, Lucia Manso-Silván**

ASTRE, CIRAD, INRAE, Univ Montpellier, Montpellier, France

\* philippe.totte@cirad.fr

**Data Availability Statement:** Data are available online at https://zenodo.org/record/7442582#.ZAiYrHbMJOC.

## Abstract

We investigated the interactions of unopsonized and opsonized *Mycoplasma mycoides* subsp. *mycoides* (*Mmm*) with bovine macrophages *in vitro*. *Mmm* survived and proliferated extracellularly on bovine macrophage cell layers in the absence of *Mmm*-specific antisera. Bovine complement used at non-bactericidal concentrations did neither have opsonizing effect nor promoted intracellular survival, whereas *Mmm*-specific antisera substantially increased phagocytosis and *Mmm* killing. A phagocytosis-independent uptake of *Mmm* by macrophages occurred at a high multiplicity of infection, also found to induce the production of TNF, and both responses were unaffected by non-bactericidal doses of bovine complement. Bovine complement used at higher doses killed *Mmm* in cell-free cultures and completely abrogated TNF responses by macrophages. These results provide a framework to identify *Mmm* antigens involved in interactions with macrophages and targeted by potentially protective antibodies and point towards a pivotal role of complement in the control of inflammatory responses in contagious bovine pleuropneumonia.

## Introduction

*Mycoplasma mycoides* subsp. *mycoides* (*Mmm*) is a wall-less bacterium and the causative agent of contagious bovine pleuropneumonia (CBPP), a highly contagious respiratory disease of cattle notifiable to the World Organization for Animal Health [1]. CBPP lung lesions and chronic carrier state are suggestive of an uncontrolled inflammatory response by the host's immune system and immune evasion mechanisms developed by *Mmm* respectively [2]. Although macrophages patrolling the surface of respiratory mucosae represent the first line of defense against pathogens invading the lung, little is known of the interactions of these cells with *Mmm*. Macrophages containing *Mmm* antigens are found in the lungs of infected animals [3] and they produce the archetypal pro-inflammatory cytokine TNF upon infection *in vitro* [4]. However, studies investigating the fate of *Mmm* in the presence of macrophages are lacking.

Another major contributor to the host defense against infection is the complement system. Complement proteins are present in extracellular fluids allowing direct killing and non-specific opsonization, followed by increased phagocytosis of pathogens [5]. Non-specific

**Funding:** The author(s) received no specific funding for this work.

**Competing interests:** The authors have declared that no competing interests exist.

opsonization can result in killing but also in intracellular survival if pathogens are able to prevent fusion of phagosomes with lysosomes [6, 7]. Upon activation, the complement promotes inflammation through the induction of pro-inflammatory cytokines [8, 9]. These pleiotropic effects of complement are not exclusive and may be dose-dependent. For example, the complement membrane attack complex responsible for the pathogen killing activity of complement can also induce the production of pro-inflammatory cytokines by phagocytes when used at non-bactericidal doses [9]. Thus, in CBPP, interactions between *Mmm*, macrophages and complement may either benefit the host, through direct or opsonization-mediated killing of *Mmm* or: paradoxically; harm the host, through opsonization-mediated intracellular survival of *Mmm* and amplification of inflammatory responses.

Although previous work suggests that mycoplasmas are generally phagocytosed by macrophages only in the presence of specific antiserum [10], the question remains open for *Mmm*. Also, effects of complement other than opsonization, such as direct killing and cytokine production, may impact disease outcome in CBPP. Therefore, the following study was undertaken to investigate the interaction of bovine macrophages with *Mmm* under physiological conditions, *i.e.*, in the presence of complement prepared from the same cell donor. Firstly, the effect of bovine complement on the viability of *Mmm* was analyzed to determine non-bactericidal concentrations. Secondly, the uptake of *Mmm* by macrophages and subsequent extra- and intracellular survival in the presence of bovine complement at non-bactericidal doses was investigated. Finally, we addressed the possibility that complement acts as a potentiator of *Mmm*-induced TNF responses of macrophages.

## Methods

### Bacterial strains

Non-fluorescent *Mmm* strain 8740-Rita and *Mycoplasma bovis* strain Oger2, and fluorescent mNeonGreen-expressing *Mmm* mutant strain 8740-Rita clone 6 (RN6), were grown in PPLO-based medium as described previously [11]. Mycoplasmas obtained from log-phase cultures were washed once in PBS by centrifugation at 10000 x *g* for 10 min before resuspension in macrophage culture medium (see below) at $1-2x10^9$ colony forming units per mL, or CFU/ml, as determined through dilution and plating on PPLO-based media supplemented with 1% Noble Agar (Difco, USA). Briefly, serial 10-fold dilutions in PBS were prepared in a final volume of 500μL with a change of tip and vortexing between each tube. 20μL from each tube were spotted onto PPLO agar plates and incubated at 37˚C for four days before counting the colonies under a light microscope.

### Bovine complement and antiserum

Non-decomplemented bovine sera obtained from 3 adult Jersey cattle housed at Cirad's animal facility served as a source of complement. These animals were purchased in France were CBPP has been eradicated and were kept outdoor after approval by the Languedoc-Roussillon regional ethics committee (French CE-LR#36) in the authorized project using animals for scientific purposesAPAFIS#27628. Sera were aseptically collected from whole blood tubes after clotting for 30 min at room temperature at 19–23˚C, followed by centrifugation at 800 x g for 20 min. Single-use aliquots of non-decomplemented sera were kept in liquid nitrogen until use after quick thawing at 37˚C. Bactericidal capacities of bovine complement were assessed by incubation of $1x10^6$-$1x10^8$ viable *Mmm* with different concentrations of non-decomplemented bovine sera at 100 μL/well in 96-well flat-bottom-plates for 1 h at 37˚C and 5% $CO_2$ before vortexing and CFU titration. That range of viable *Mmm* corresponds to the lowest and highest number of bacteria used to infect macrophages. The specific *Mmm* antiserum was obtained by

pooling sera from two-years old Zebu cattle recovering from an experimental infection with *Mmm* strain 8740-Rita [12]. These animals were selected on the basis of their positivity using a CBPP-specific cELISA [13] and a record of clinical signs typical of CBPP. Prior to use, *Mmm* antisera were decomplemented by heating at 56˚C for 30min.

## Macrophages and bacterial infection

Macrophages were derived from monocytes obtained from the same three animals used to prepare non-decomplemented bovine sera. Briefly, monocytes were purified from PBMC by positive selection of CD14+ cells using anti-human CD14 magnetic beads (Miltenyi, Germany) according to the manufacturer's instructions. Monocytes were resuspended in Iscove's Modified Dulbecco's Medium (IMDM) supplemented with 2 mM L-glutamine, 50 μM 2-mercaptoethanol, 0.4 mg/mL ampicillin and 10% heat-inactivated fetal calf serum (FCS; Eurobio AbCys, France), seeded at $3 \times 10^5$ cells/well of 96-well flat-bottom-plates in 200 μl, and incubated for 6 days at 37˚C and 5% $CO_2$ with a change of half of the medium at day 3. Before infection, the medium was removed and replaced with IMDM lacking FCS to which the following stimuli were added to reach a final volume of 100 μl/well: i) appropriate amounts of mycoplasmas to obtain MOIs of 1, 10 and 100; and ii) non-decomplemented autologous bovine sera or antiserum or FCS. Non-decomplemented bovine sera were used at non-bactericidal doses ranging from 5 to 10% v/v depending on the donor, whereas antiserum was used at 20% v/v and FCS at 10% v/v. After 1 h incubation, the medium was replaced and 100μl of complete IMDM medium was added to the wells before harvesting or further incubation. For harvesting macrophages, monolayers were scrapped off using 200μl tips, pelleted by centrifugation at 500 x g for 5 min, resuspended in 200 μL PBS, and passaged 5 times through a 27gauge needle, which lysed more than 80% of cells as shown by Trypan blue viability counts. *Mmm* CFU titers were determined as described above. For gentamycin assays, an MOI of 10 was used, as it was shown in preliminary macrophage-free assays that addition of 400 μg/ml of gentamicin to the culture for 3 hours reliably killed 100% of mycoplasmas. Cultures infected with *M. bovis* strain Oger2 were used as positive controls of intracellular survival in the presence of gentamycin [14].

## Flow cytometry

Macrophages were infected with fluorescent and non-fluorescent *Mmm* at MOIs of 500–1000 as described above. Due to the high MOI, non-decomplemented bovine sera could be used at up to 40% v/v without affecting *Mmm* viability. Before harvesting, the medium was replaced by 100 μl/well of fresh medium and cells were scraped off, homogenized by pipetting up and down, and transferred to 5 ml cytometry tubes. After addition of 10 μL/tube of propidium iodide (PI, BD Biosciences, USA) and 10min incubation at 4˚C in the dark, 100 μL of sheath fluid was added to each tube. Preliminary macrophage-free tests showed that 100% of *Mmm* took up PI in these conditions. Since PI and mNeonGreen fluoresce in different wavelength, it was possible to gate out PI positive macrophages to exclude dead cells (i.e., with loss of membrane integrity) and extracellular *Mmm*. This allowed analysis of strictly intracellular mNeonGreen-dependent fluorescence. CytochalasinD (Sigma, France), an inhibitor of actin-dependent phagocytosis, was used at 10 μg/ml. At least 5000 events were analyzed within the PI negative gate with a FacsCanto flow cytometer (BD Biosciences) equipped with the FacsDiva software (BD Biosciences). Gates and quadrants were used to delineate populations of interest and calculate statistics respectively. Results were expressed in percentages of cells fluorescing above background (*i.e.*, fluorescence produced by macrophages infected with non-

fluorescent *Mmm*) and geometric mean fluorescence intensity (MFI), which is proportionally related to numbers of mNeonGreen+ *Mmm* per cell.

### ELISA

Supernatants were collected from macrophage cultures infected as above for 24 h with *Mmm* at MOIs ranging from 100 to 1000 and stored at -20˚C until use. Non-decomplemented bovine sera were used at both bactericidal and non-bactericidal concentrations and the highest dose was also used for FCS. Lipopolysaccharide (LPS) from *Escherichia coli* 0111:B4 (Invivogen, France), used at 2 μg/ml, served as a positive control. To quantify bovine TNF, a sandwich ELISA was performed using a matched antibody pair (BioRad, France) as previously described [15]. Supernatants were tested in duplicates and results obtained from a labsystems multiskan MS ELISA reader were expressed in mean optical densities.

### Statistics

A non-parametric Mann–Whitney U-test (https://www.socscistatistics.com/tests/mannwhitney/) was used to analyze differences between responses obtained with various stimuli. Single and double asterisks represent *p* values of <0.05 and <0.01 respectively. A difference was considered to be significant at a *p* value of <0.05.

## Results

### Bactericidal effect of bovine complement on *Mmm* viability in the absence of macrophages

A dose-dependent effect of non-decomplemented bovine sera, containing bovine complement, on *Mmm* CFU titers was observed *in vitro* (Fig 1A). Depending on the dose and animals used the decrease in CFU titers varied between 0 and 3 logs, indicating a potent effect of bovine sera and a high variability among animals. There was no sign of *Mmm* aggregation induced by non-decomplemented serum when cultures were checked under a light microscope, and passage through a 27G needle had no impact on CFU titers, indicating that reduced viability occurred through direct cytotoxicity. The killing effect was due to complement since it was neutralized by heating sera at 56˚C for 30 min (Fig 1B) a method known to inactivate complement [5]. The mycoplasmacidal activity and inter-animal variability of bovine sera remained consistent throughout the study (S1 Fig).

### Effect of bovine complement on *Mmm* opsonization and survival in the presence of macrophages

A protocol was used to focus on *Mmm* associated to macrophages and to exclude non-adherent and plastic-adherent mycoplasmas from the analysis (see methods). Furthermore, no cytopathological effects of *Mmm* on macrophage monolayers were detected at any MOI used. In the absence of bovine complement and antiserum, complete survival of *Mmm* occurred at 24 h and growth of up to 1 log was observed at 48h, indicating no killing activity by macrophages (Fig 2). Similar trends were observed at MOIs of 1 and 100 whereas only survival of *Mmm* without growth (i.e., CFU titers remained constant) was observed in the absence of macrophages (S2 Fig). The addition of non-decomplemented bovine sera at non-bactericidal concentrations had no effect on *Mmm* viability at any time-points (Fig 2). As expected, addition of *Mmm*-specific antiserum induced significant killing activity after 24 h and up to 2.5 logs decrease in CFU titers at 48h (Fig 2).

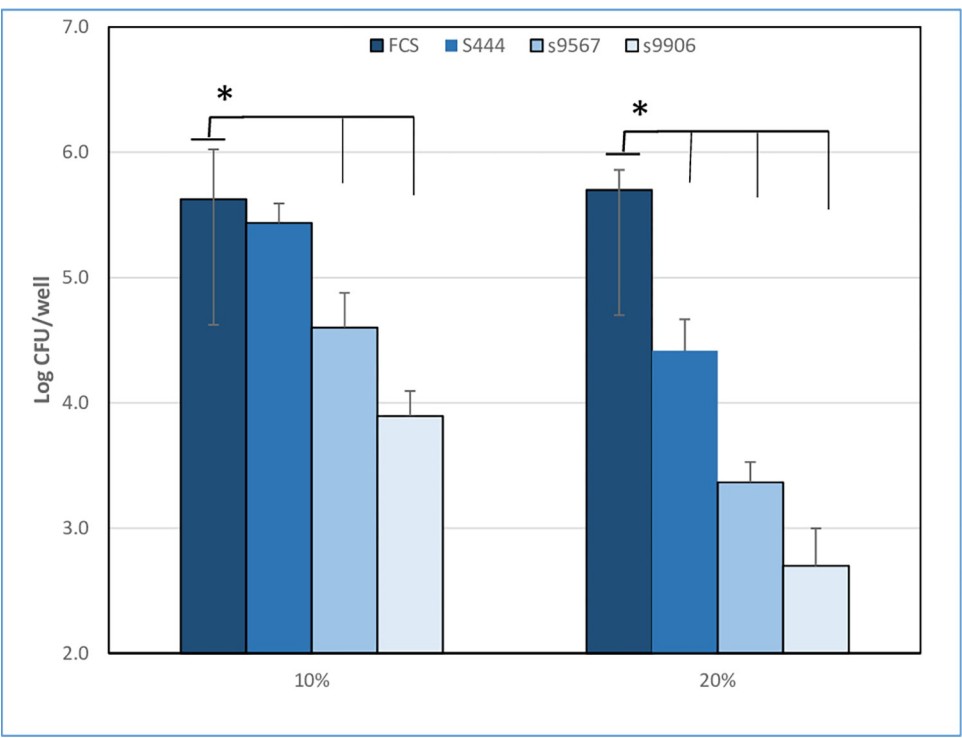

**Fig 1.** Cytotoxic effect of bovine complement on *Mmm* in vitro before a) and after b) heating at 56˚C. *Mmm* was incubated for 1 h with decomplemented sera (FCS) or different concentrations (10%, 20%) of bovine sera from 3 different animals (s+animal number). Results from 3 different experiments are shown as mean+/-SD of CFU expressed in log10. Asterisks and bars indicate statistically significant differences with FCS.

In order to analyze intracellular survival of *Mmm* experiments were repeated in the presence of gentamicin to kill all extracellular mycoplasmas. This was performed only at an MOI of 10, which represented a concentration of $3 \times 10^6$ CFU/ml of *Mmm*, since gentamicin did not kill 100% extracellular mycoplasmas at higher *Mmm* concentrations. As seen in Fig 3, gentamicin assays indicated that the overwhelming majority of *Mmm* survival at 24 h was due to extracellular mycoplasmas and that bovine complement had no effect on *Mmm* survival inside macrophages. This was not due to a bias in gentamicin assay since significant survival was observed with *M. bovis* Oger2 (Fig 3).

### Effect of bovine complement on the uptake of fluorescent *Mmm* by macrophages

We were able to analyze *Mmm*-dependent intracellular fluorescence by gating out extracellular fluorescence as explained in "Methods". As seen in Fig 4, in the absence of bovine complement or antiserum, more than 90% of macrophages had engulfed fluorescent *Mmm* but at low levels as indicated by low FITC fluorescence intensity ("Li" in Fig 4A). Indeed, FITC signals did not exceed one log above background level and could not be detected when MOIs of 100 or less were used (data not shown). No differences were detected between 1 and 24 h PI in both percentages of positive cells and MFI (Fig 4A and 4B). Non-decomplemented bovine sera used at non-bactericidal concentrations had no effect on FITC fluorescent levels at any time-points (Fig 4A and 4B). Data at 48h were not exploitable due to high background fluorescence. The low-level uptake was not affected by cytochalasinD ("Li" in Fig 4C) indicating that it was not dependent on phagocytosis. A substantially stronger FITC fluorescence intensity was observed

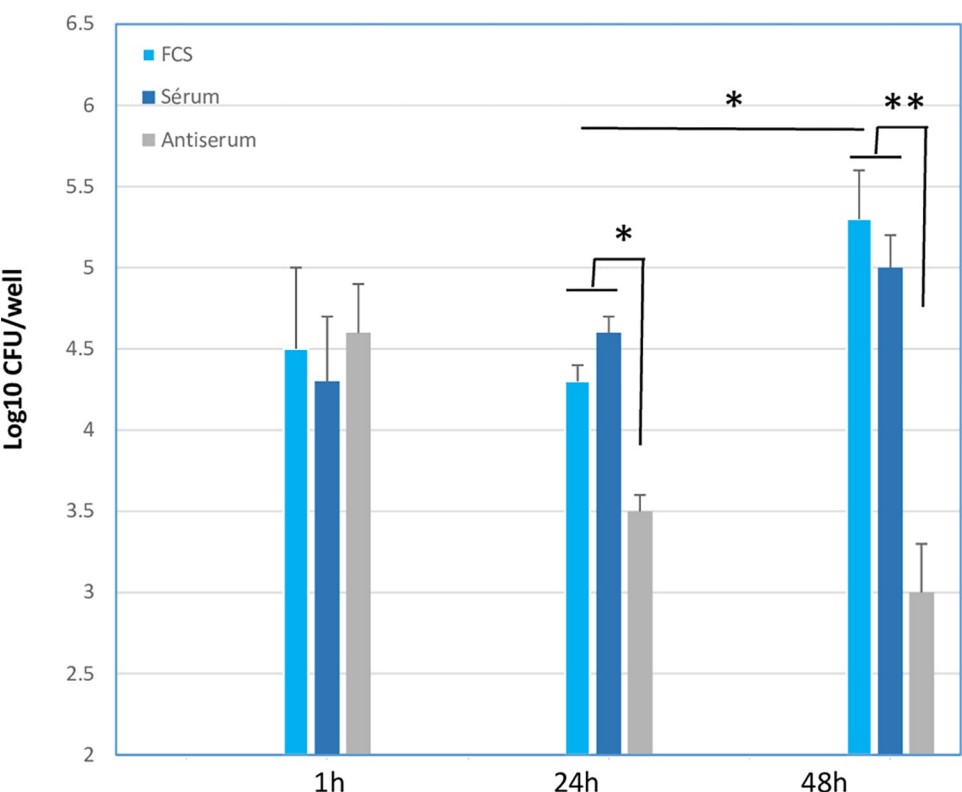

**Fig 2. Macrophages do not kill *Mmm* even in the presence of bovine complement.** Macrophages were infected with *Mmm* at an MOI of 10 in the absence (FCS) or presence (serum) of non-decomplemented bovine sera at non-bactericidal concentrations and incubated for 1, 24, and 48h before assessment of macrophages-associated *Mmm* titers. Anti-*Mmm* antiserum (antiserum) was added to additional wells as positive controls of bactericidal activity. Results from 3 different experiments using 3 animals are shown as mean+/-SD of CFU expressed in log10. Asterisks indicate statistically significant differences with FCS.

in the presence of antiserum indicating a strong increase in *Mmm* uptake by macrophages ("Hi" in Fig 4A). At 24 h PI, the % of cells within the "Hi" gate increased while the mean fluorescence intensity decreased (gate P5 and MFI in Fig 4B) suggesting partial destruction of fluorescent *Mmm* within macrophages. Moreover, the vast majority of highly fluorescent cells disappeared in the presence of cytochalasinD (Fig 4C) indicating that the uptake of *Mmm* by macrophages in the presence of antiserum was dependent on phagocytosis.

The lack of effect by bovine complement on the uptake of *Mmm* by macrophages was confirmed for all three animals tested (Table 1). Indeed, the addition of non-decomplemented bovine sera at non-bactericidal concentrations had no impact on numbers of positive cells and MFI when compared to FCS at 1 h and 24 h PI. In addition, the positive effect of antiserum on *Mmm* uptake was confirmed as well as the increase in percentage of positive cells at 24 h PI accompanied by a decrease in MFI, indicating progressive destruction of phagocytized fluorescent *Mmm*.

## Effect of bovine complement on *Mmm*-induced TNF production by macrophages

As shown in Fig 5, in the absence of bovine sera and antiserum, *Mmm* induced the production of TNF by macrophages when MOIs of 500–1000 were used. At an MOI of 100, TNF was not detected (not shown). There was no effect of bovine complement at non-bactericidal doses

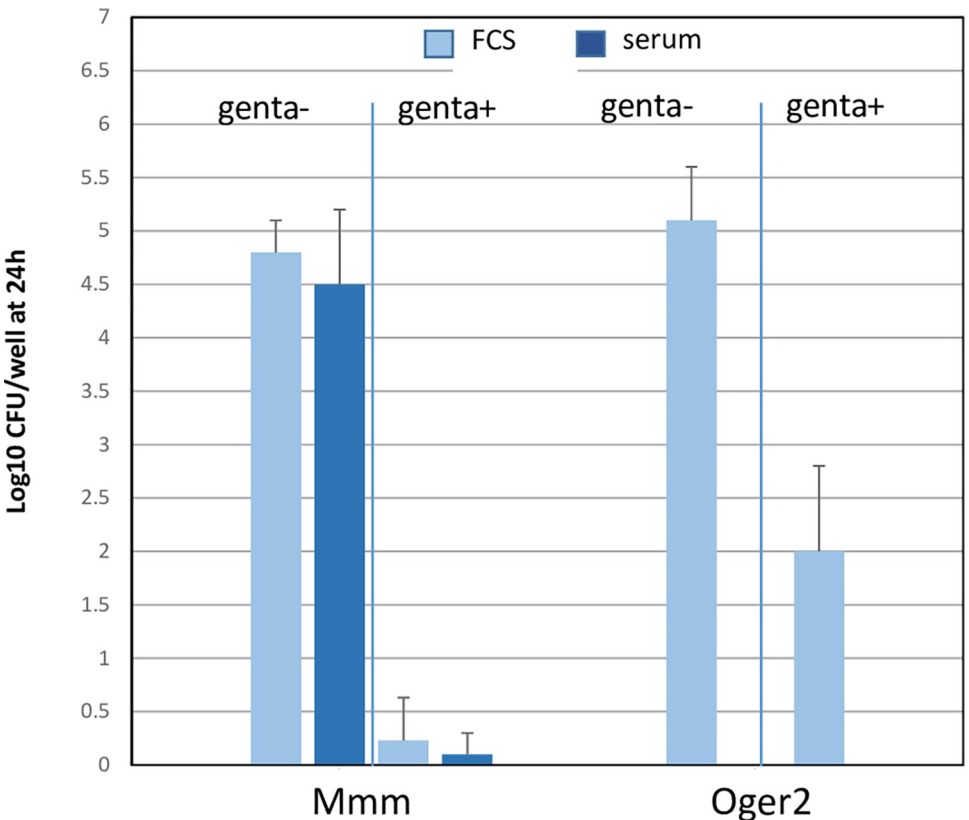

**Fig 3. Bovine complement has no impact on *Mmm* survival inside macrophages.** Macrophages were infected with *Mmm* at an MOI of 10 in the absence (FCS) or presence (serum) of bovine complement at non-bactericidal concentrations and incubated for 24 h. Cells were then treated for 4-hours with gentamycin before harvesting and assessment of macrophage-associated *Mmm* titers. Wells containing macrophages infected with *M. bovis* (Oger2) were used as positive controls of intracellular survival in the presence of gentamycin. Results from 3 different experiments using 3 animals are shown as mean+/-SD of CFU expressed in log10.

(Fig 5, serum1). In order to analyze the effect of non-decomplemented bovine sera at bactericidal doses, concentrations of both non-decomplemented bovine sera and FCS had to be increased. For one animal, a substantial increase in background (*i.e.*, TNF induced by FCS alone in the absence of *Mmm*) prevented its use. Nevertheless, non-decomplemented bovine sera from the remaining two animals, used at bactericidal concentrations inducing at least 1log decrease in *Mmm* viability, almost completely abrogated the production of TNF induced by *Mmm* (Fig 5, serum2).

## Discussion

Given their pivotal role in first-line immune defenses against mucosal pathogens, we have investigated the interactions between macrophages and *Mmm*, the causal agent of CBPP. Non-decomplemented bovine sera were used as a source of complement to mimic physiological conditions present at infected sites and cryopreservation in liquid nitrogen was instrumental in overcoming the instability of complement. The strong bactericidal effect of some non-decomplemented bovine sera on *Mmm* viability that we observed in this study is in conflict with previous work [16]. One likely explanation is that we used bovine sera instead of guinea pig sera. Similarly, non-decomplemented caprine sera was found to reduce the viability of *Mycoplasma mycoides* subsp. *capri* (*Mmc*), *Mmm*'s closest relative [17]. Given its stability over

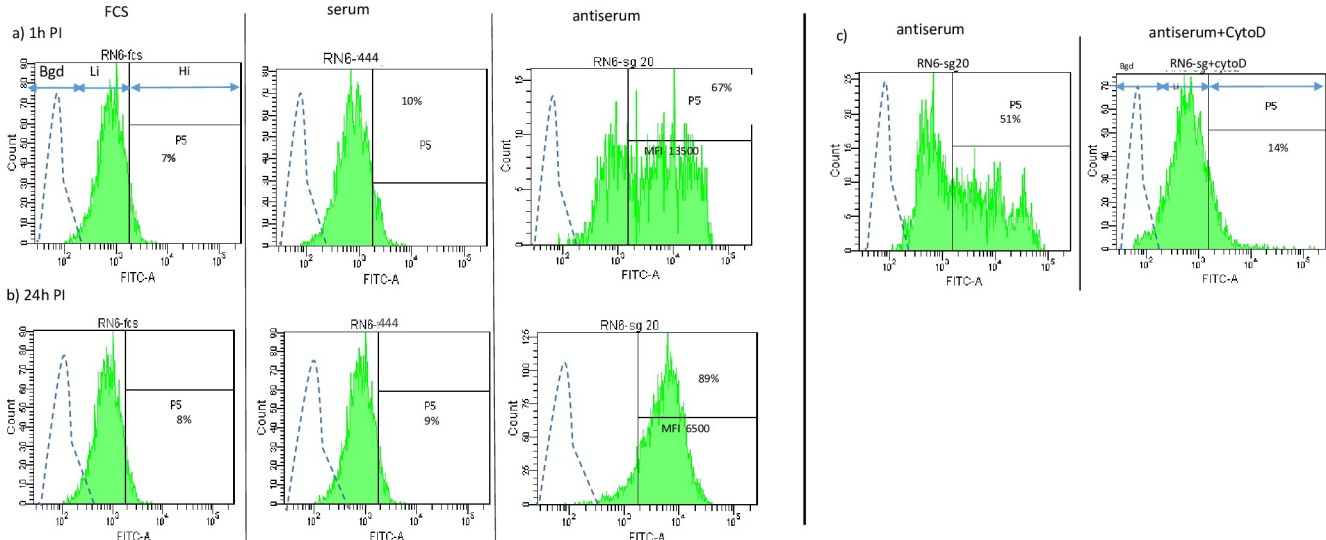

**Fig 4. Bovine complement does not impact *Mmm* uptake by macrophages.** Macrophages were infected with either non-fluorescent or fluorescent *Mmm* at an MOI of 500–1000, and either in the absence (FCS) or in the presence (serum) of non-decomplemented bovine sera at non-bactericidal concentrations. They were incubated for 1 h (a) and 24 h (b) before analysis of intracellular FITC fluorescence by flow cytometry (see Methods). Finally, the effect of cytochalasinD, an inhibitor of phagocytosis, on *Mmm* uptake by macrophages is shown in c). Anti-*Mmm* antiserum (antiserum) was added to additional wells as positive control of specific phagocytosis. Typical histogram plots are shown for one representative animal as an example. Histograms and arrows display background (bgd, hatched line), and low (Li, green line) and high (Hi, green line) fluorescence intensities produced by macrophages infected with non-fluorescent and fluorescent *Mmm* respectively. Percentages of highly fluorescent cells and MFI are given by the P5 interval.

time *in vivo* and variable potency between animals, it would be interesting to assess whether a correlation exists between the anti-*Mmm* effect of bovine sera and the resistance of animals to CBPP. Considering the pleiotropic effects of complement on immune responses to pathogens, we have explored the possible impact of bovine complement, used at non-bactericidal concentrations, on important functions of macrophages such as non-specific opsonization-mediated phagocytosis and production of pro-inflammatory TNF.

In the first place, we analyzed the kinetics of *Mmm* survival in the presence of bovine macrophages and in the absence of complement and antiserum. Lysis of macrophages prior to *Mmm* titration allowed taking into account both intracellular and membrane-associated

**Table 1. Effect of bovine complement (serum) from three different animals (mean+/-SD) on the uptake of fluorescent *Mmm* by autologous macrophages as measured by flow cytometry 1 h and 24 h post infection.**

| Stimuli | Low fluorescence intensity[a] | | | | High fluorescence intensity[a] | | | |
|---|---|---|---|---|---|---|---|---|
| | % pos [b] | | MFI[I] | | % pos [b] | | MFI[c] | |
| | 1 h | 24 h | 1 h | 24 h | 1 h | 24 h | 1 h | 24 h |
| FCS | 93+/-2 | 96+/-5 | 1013+/-33 | 995+/-33 | 7+/-3 | 10+/-1 | NA | NA |
| Serum | 92+/-3 | 99+/-3 | 1008+/-52 | 1050+/-42 | 8+/-4 | 9+/-2 | NA | NA |
| Antiserum[d] | NA | NA | NA | NA | 63+/-10 | 94+/-5 | 14518 | 6905 |
| | | | | | | | +/-1313 | +/-948 |

(a): low and high fluorescence intensities above background fluorescence (see also Fig 4)

(b): Percentages of positive celI(c): Mean fluorescence intensi ty

(d): sera from CBPP convalescent animals

NA: not applicable. The effect of antiserum is measurable only within the high fluorescence intensity whereas, in the presence of complement, the vast majority of positive cells were found in the low fluorescence intensity range.

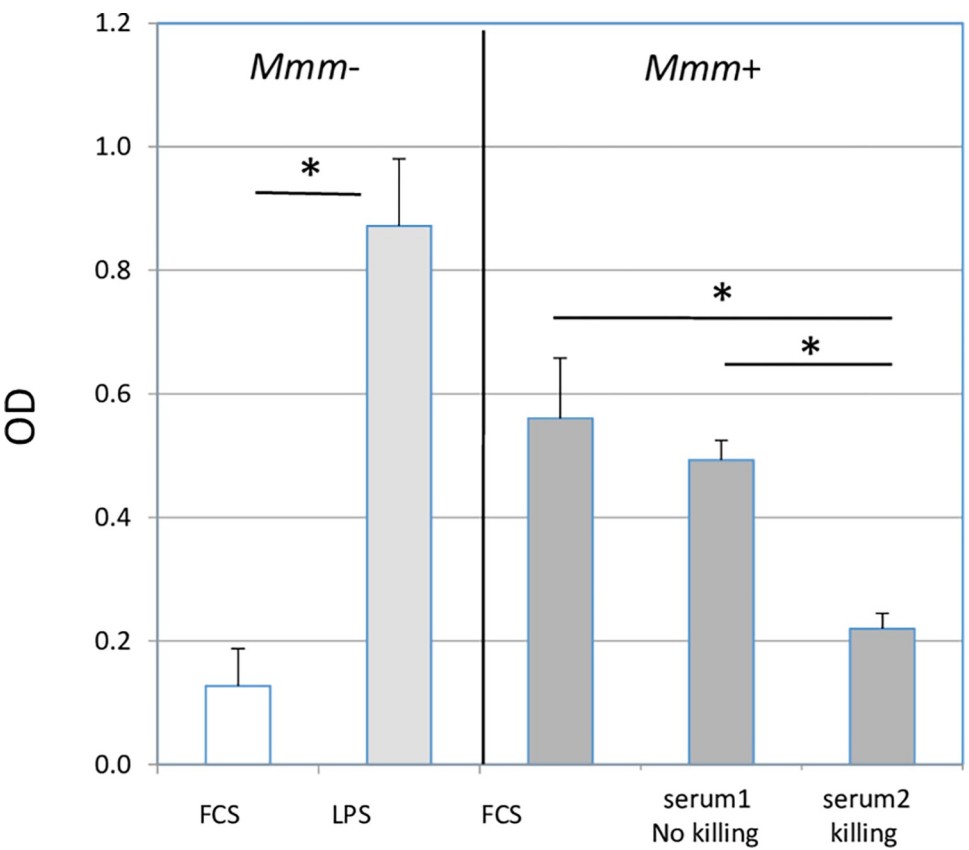

**Fig 5. Effect of bovine complement on *Mmm*-induced release of TNF by autologous macrophages.** Macrophages were either left uninfected (*Mmm-*) or infected (*Mmm+*) with *Mmm* at an MOI of 500–1000 in either the absence (FCS) or presence of non-decomplemented bovine sera at non-bactericidal (serum1) and bactericidal (serum2) concentrations and incubated for 24 h before TNF titration of supernatants. Lipopolysaccharide (LPS) was added to additional wells as positive controls. Results from 3 different experiments using complement from two animals are shown as mean+/-SD of optical densities (OD). Asterisks indicate statistically significant differences.

mycoplasmas. *Mmm* CFU titers remained stable after 24 h and increased by up to 1log after 48h, indicating the absence of measurable killing by macrophages. Little or negligible numbers of macrophage-associated *Mmm* survived intracellularly in the presence of gentamicin, suggesting that viable *Mmm* were mainly located on cell membranes. Absence of killing by macrophages or even growth on macrophages in the absence of specific antiserum and complement has been reported previously for other mycoplasmas [10, 18]. We found no measurable effect of bovine complement on *Mmm* CFU titers, suggesting that neither non-specific opsonization-mediated killing nor intracellular survival occurred under our experimental conditions. It should be noted that pre-incubation of *Mmm* with bovine complement for 30 min at 37˚C before infecting macrophages rather than concomitantly did not change the results. However, we cannot exclude the possibility that the presence of *Mmm* growing on macrophages masks a low-level killing activity. On the other hand, *Mmm* titers substantially decreased over time when anti-*Mmm* antiserum was added to the medium indicating that intrinsic killing capacities of macrophages were not altered. Again, absence of killing by macrophages, even in the presence of complement, has been shown previously for several mycoplasmas [10]. It is possible that the presence of a polysaccharide capsule [19] protects non-opsonized and non-specifically opsonized *Mmm* from phagocytosis [20] which may be further investigated using non capsulated variants [16].

In order to gain a better insight on the interactions of *Mmm* with bovine macrophages, we used a more direct approach based on flow cytometry-assisted monitoring of intracellular fluorescence of phagocytes after infection with fluorescent *Mmm*. Our results show that, in the absence of complement and antiserum, more than 90% of macrophages take up fluorescent *Mmm* at low levels, as indicated by low mean fluorescence intensities. Unfortunately, in macrophages-free cultures, gentamicin was not 100% bactericidal at the *Mmm* concentrations used for flow cytometry.Thus, preventing assessment of the viability of intracellular fluorescent *Mmm*. The lack of decrease in fluorescence over time suggests that either no killing occurred or that it was not sufficient to cope with ongoing growth of *Mmm* on macrophages. CytochalasinD had no effect on the number of cells harboring low intracellular fluorescence, suggesting that the uptake of *Mmm* by macrophages was independent of phagocytosis. Interestingly, it has recently been shown that *Mmc* interacts with C-type lectin receptors (CTLRs) [21] some of which mediate endocytosis. Several CTLRs are present on macrophages, which warrants further studies on their potential role in CBPP. Moreover, pathogen-associated molecular patterns (PAMPs) of *Mmm* recognized by these receptors have potential as virulent factors. Bovine complement used at non-bactericidal concentrations had no effect on the uptake of *Mmm* by macrophages at any time points, confirming viability experiments using CFU titration. On the other hand, anti-*Mmm* antiserum induced a strong and phagocytosis-dependent increase in *Mmm* uptake leading to more than 90% of cells located in the high fluorescence intensity gate at 24 hpi. Moreover, the mean fluorescence intensity within that gate decreased at 24 hpi confirming the progressive destruction of *Mmm* as seen with CFU titration. These last results are intriguing since *Mmm* and related mycoplasmas possess a two-protein system capable of cleaving immunoglobulins G (IgG) [22]. Decomplementation of anti-*Mmm* antisera did not affect their opsonizing potency indicating that IgGs are most likely involved.

Our results confirm previous work demonstrating the capacity of *Mmm* to induce TNF production by alveolar macrophages in the absence of either complement or antiserum (Jungi et al., 1996). Our results show that the use of peripheral blood-derived MDMs represents a valid alternative to alveolar macrophages, whose preparation requires greater expertise and has a greater impact on animal welfare. Interestingly, TNF was induced only at high MOIs (i.e., >500) also shown by flow cytometry to be associated with phagocytosis-independent engulfment of *Mmm*. This suggests a possible involvement of autophagy in the pro-inflammatory response of macrophages to *Mmm*. There was no beneficial effect of bovine complement on the TNF response of infected macrophages when used at non-bactericidal concentrations. The effect of bovine complement at bactericidal doses was also analyzed, considering the possible release of pro-inflammatory molecules from *Mmm* undergoing lysis. We found that, when used at bactericidal concentrations, bovine complement efficiently abrogated the production of TNF by macrophages in response to *Mmm* infection.

In summary, using a combination of CFU titration, gentamicin assays, fluorescent *Mmm*, and TNF ELISA, we were able to show that: i) macrophages were not capable of killing *Mmm* efficiently in the absence of antiserum; ii) bovine complement used at non-bactericidal doses neither had opsonizing effect nor did it promote intracellular survival; iii) bovine complement did not impact the phagocytosis-independent *Mmm* uptake by macrophages that occurred in the absence of antiserum; iv) bovine complement used at non bactericidal concentrations did not amplify *Mmm*-induced TNF responses of macrophages, and even abrogated that response when used at concentrations inducing bactericidal activity. These results provide a framework to identify PAMPs as well as *Mmm* antigens recognized by CTLRs and protective antibodies. In addition, they point towards a role for complement in the control of TNF-dependent inflammatory responses in CBPP.

## Supporting information

**S1 Fig. Stability of the bactericidal effect of bovine complement.**
(TIF)

**S2 Fig. Survival of *Mmm in vitro* in supplemented IMDM medium in the absence of macrophages.**
(TIF)

## Author Contributions

**Conceptualization:** Philippe Totté, Lucia Manso-Silván.

**Funding acquisition:** Lucia Manso-Silván.

**Investigation:** Philippe Totté.

**Methodology:** Philippe Totté, Tiffany Bonnefois, Lucia Manso-Silván.

**Validation:** Philippe Totté.

**Writing – original draft:** Philippe Totté.

**Writing – review & editing:** Philippe Totté, Lucia Manso-Silván.

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
