## [Decision Letter · Decision Letter 0]

10 Jan 2024

PONE-D-23-32412Interactions between Mycoplasma mycoides subsp. mycoides and bovine macrophages under physiological conditionsPLOS ONE

Dear Dr. Totté,

Thank you for submitting your manuscript to PLOS ONE. After careful consideration, we feel that it has merit but does not fully meet PLOS ONE’s publication criteria as it currently stands. Therefore, we invite you to submit a revised version of the manuscript that addresses the points raised during the review process.

We look forward to receiving your revised manuscript.

Kind regards,

Jianhong Zhou

Staff Editor

PLOS ONE

2. We noted in your submission details that a portion of your manuscript may have been presented or published elsewhere. [the article was preprinted and recommended by PCI https://microbiol.peercommunityin.org/articles/rec?id=5] Please clarify whether publication was peer-reviewed and formally published. If this work was previously peer-reviewed and published, in the cover letter please provide the reason that this work does not constitute dual publication and should be included in the current manuscript.

Reviewers' comments:

Reviewer's Responses to Questions

**Comments to the Author**

1. Is the manuscript technically sound, and do the data support the conclusions?

Reviewer #1: No

Reviewer #2: Yes

2. Has the statistical analysis been performed appropriately and rigorously? 

Reviewer #1: No

Reviewer #2: Yes

3. Have the authors made all data underlying the findings in their manuscript fully available?

Reviewer #1: Yes

Reviewer #2: No

4. Is the manuscript presented in an intelligible fashion and written in standard English?

Reviewer #1: No

Reviewer #2: Yes

5. Review Comments to the Author

Reviewer #1: Comments to authors

In the manuscript by Totté et al., the authors have described interactions of Mycoplasma mycoides subsp. mycoides (Mmm) with bovine macrophages using in vitro killing and phagocytosis assays. While the manuscript focuses on an essential area of research of Contagious Bovine Pleuropneumonia by conducting experiments to identify interaction of macrophages (from the target species) with Mmm, the data reported in the manuscript is not enough to support the current conclusions. Moreover, the authors have made various highly speculative statements without providing any evidence whatsoever to in the manuscript.

Unfortunately, in its current form the manuscript cannot be accepted. The authors must provide further experimental data in order to consider the publication of the manuscript. These are mentioned below.

Line 108-109: In this manuscript, the methodology used to study the interaction of Mmm and macrophages is extremely important. So, Mmms were left with the macrophages for 1 hr in all the experiments? Were the cells washed before adding different serums.

The way methodology is written currently, it seems that both mycoplasmas and serums were added together. It is very important for the results observed in Fig 2 because if the mycoplasmas and serums were added together then its really not different compared to the data presented in Fig 1.

The authors need to clarify this.

Flow cytometry analysis (Line 130-132): The flow cytometry analyses are important in order to understand whether Mmms are internalised (gone inside of the macrophages) or associated (made complex and stuck outside) with macrophages. Based-on the current gating strategy how would you differentiate macrophage-associated Mmm from macrophage-internalised Mmm?

Given that Mmms are so small, it will be very challenging to differentiate the macrophage-associated Mmm from macrophage-internalised Mmm. Due to this limitation, both the externally bound Mmm (large proportion) and internalised (minimum proportion) to macrophages will generate the same double positive signal (FITC + PI). Therefore, to understand what is going on here the side scatter and forward scatter graphs must be presented. In addition, the gating strategy must be presented in the supplementary information.

Line 180: Based on the methodology, it doesn't see that the authors have focused on Mmm associated to the macrophages. Please clarify.

Line 181: The authors mention that "no cytopathological effects were detected on macrophages". Did the authors measure viability of the macrophages?

Line 186: This is bit confusing. MOI is always represented in relation to cells, but the authors are saying in the absence of macrophages. Please clarify. The data must be presented for MOI 1 and 100 even thought, these MOI showed a similar trend.

Line 205: How was this conclusion made? The authors just mentioned that that "overwhelming majority of Mmm survival at 24 h was due to extracellular mycoplasmas". I don't think that the authors can make this conclusion based on the current results. First, it needs to be demonstrated that Mmm can survive in macrophages. Given that very low number of Mmm were detected in the G+ treatment group, it is almost impossible to say that "complement had no effect on Mmm survival inside macrophages".

Line 212: Fig 3 Given that the authors performed experiments to assess intracellular survival of Mmm in macrophages, this should be macrophages-internalised and not macrophages-associated.

Line 220: These results are contradictory to the previous observations presented in Fig 3. In that, hardly any Mms were internalised but here the authors claim that over 90% were engulfed by the macrophages. Please clarify these contradicting results.

Also, proper gating strategy must be presented in the supplementary information including side and forward plots, gating for any possible clumps of neogreen mutant strain.

What was the rational of using 500-1000 MOI of Mmm in these experiments? In the previous Fig. the authors have mentioned that gentamicin did not kill Mmm at MOI higher than 100, indicating that Mmm higher than MOI of 100 are likely to associated with macrophages and NOT internalised. Please describe these discrepancies in your results.

Line 222: While it is understandable the rational of using neogreen expressing strain, I think there is bigger fundamental problem here. Given that no FITC signal was detected when MOI of 100 or less were used, it clearly indicates that both the WT and neogreen expressing strains are not behaving the same when it comes to their ability to infect or engulfed by macrophages. The first question, which must be answered that how comparable these two different strains are in terms of their ability to internalise or engulfed by macrophages. As far as I see, there is no data on this. Therefore, no conclusion can be made from any comparisons of the mutant and WT Mmm strains in this study.

Line 226: Given that the uptake (internalisation) is low, this conclusion cannot be made.

Line 242: Fig 4 By looking at the graphs, background fluorescence levels (hatched line) appear to be changing. Moreover, it should be at "zero" and not 10^2 or 10^3. It seems like that flow cytometry gating wasn't done properly and the gating strategy must be presented as the supplementary material.

Line 275: Mmm were diluted in PBS. Can the authors provide the source of PBS and other reagents used in the cell-culture experiments? Given that TNF levels can be easily induced even by a small amount of co-contaminating endotoxin levels in reagents, it is absolutely important to use certified endotoxin free reagents in cell culture experiments.

Line 278-279: I don't think, that is the correct way of reporting experimental findings. That's the variability of the outbred animals. You can't just take two best readings out of three. The results from the third animal must be reported. Otherwise, this experiment becomes, redundant.

Line 292: Fig 5 so, 2 animals each for serum 1 and serum 2? Is that correct?

Line 315: Yes, this is a most likely scenario, but authors kept saying that Mmms have internalised. So, results need to be updated.

Line 319: The authors should avoid using "internalisation or intracellular" because based on the results presented these conclusions can't be made.

The current flow cytometry analyses are not sufficient to conclude that Mmms are internalised into macrophages. Other methodology such as confocal microscopy could be used to analyse internalisation of Mmm into macrophages.

Line 333: Please see the comment for flow cytometry analysis.

Line 338: Again, please see the comment for line 220.

Line 345: But the authors just mentioned (in line 339) that the uptake of Mmm was independent of phagocytosis. Please clarify and discuss these in the discussion section.

Line 346: Again, the intensity can also increase if Mmm + macrophages form a complex and not necessarily internalised. So, the gating strategy must be presented with side and forward scatter graphs.

Line 352: Why did the authors not comment on the third animal, which showed substantial increase in TNF levels. The impact of that on the study should be discussed here.

Line 356: There are no evidence provided in the manuscript to support this. This statement is purely speculative. The authors should avoid making these types of statements without having any supportive data.

Line 362-368: All the above-mentioned issues must be addressed before making these i-iv conclusions. In it’s current form, the provided is not sufficient to make these conclusions.

Line 369-370: NO. This statement is totally speculative and should be rephrased.

Reviewer #2: Comments:

Abstract:

Its better to mention monocyte-derived macrophages (MDMs) in the abstract (line: 14, 21, 22, and rest of the manuscript). Because authors used MDMs rather than macrophages isolated from lung lavage and/or peritoneal cavity for the phagocytosis assay. It would be more informative to mention authors measured TNF-alpha by ELISA (abstract and other places) rather than just typing TNF as it has been mentioned in Tette et al., 2015 publication.

Materials and Methods:

Line 74: Define PBS

Line 75: change , or CFU/mL to (CFU/mL),

Line 76: define PPLO

Line 78: start the sentence with Twenty uL (not 20uL), were the tubes prepared in triplicate and/or spotted three 20uL on the plate?

Line 83: this author is not familiar with the term “non-decomplemented bovine sera” but rather familiar with “non-heat-inactivated sera”.

Line 88: remove at 19-23C

Line 93-94: change “The specific Mmm antiserum” to “The Mmm specific antiserum”

Line 97: this authors usually writes “complement was inactivated by heating….” rather than “decomplemented ….”

Line 99: change Macrophages in the title to “Monocyte-derived macrophages”

Line 109: define MOI and also typically MOI is written as 1:1; 1:10; or 1:100. This author typically write as below “multiplicity of infection (MOI) at a 1:10 ratio (monocytes: bacteria)”

Line 111: did the authors use any other percentage of sera for the assays such as 50% v/v and 90% v/v etc.

Line 114: why did not authors use Trypsin/EDTA or cold EDTA to lift the cells?

Line 127: what is the final concentration of PI? (6 ug/mL?)

Line 135: authors could define excitation and emission wave lengths for GFP and PI and the band width of filters used.

Line 146: TNF-alpha?

Line 152-155: its better to describe here how the data were presented such as mean and standard deviations (SD) [of CFU or optical densities (OD)] etc.

Results:

Line 165: is direct cytotoxicity of complement on Mmm due to the formation of membrane attack complex through alternative and/or C-type lectin activation complement pathway?

Line 166: since serum heat inactivation condition is described under methods (line 97), 56C, 30 can be removed and modify the sentence accordingly. Move the reference to line 97.

Line 176 and in other figures, just indicate * and P value since authors have already defined significance under the statistics section (line 154-155).

Line 182-184: in Fig. 1a, authors have clearly demonstrated significant killing activity with 10% serum samples from two of the three animals during 1 hr incubation. It is not clear to this reviewer then why the 10% serum had no effect when in the Fig. 2. Also why did authors used 5-10% serum vs 20% Mmm-specific antiserum knowing that 20% non-decomplemented serum can kill Mmm unless antisera were heat-inactivated (but this reviewer could not find this information from the method).

Line 189: are you suggesting here killing of Mmm with specific antisera are due to the activation of classical compliment pathway (and/or Fc-receptor mediated phagocytosis)?

Line 205-206: it was a good idea to use M. bovis as a control for gentamicin protection/killing assay.

Line 217-239: was the Mmm-specific antisera heat inactivated? If so, this phagocytosis most likely through Fc-receptor pathway.

Line 273-282: it’s better to mention authors measured TNF-alpha here because mAb used were TNF-alpha specific (Totte et a., 2015 study)

Line 273-282: the lack of TNF-alpha at lower MOI is intriguing since Sacchini et al., 2012 indicated the presence of several cytokines including TNF-alpha when cattle were challenged with Mmm. Does this mean TNF-alpha is produced by other type of cells under lower MOI (such as B-cells, NK cells, endothelia cells, fibroblast etc.?

Discussion:

Overall, nicely written discussion.

In general, the lack of phagocytosis in the presence of serum compliment is an interesting observation. Macrophages typically have all the complement receptors such as CR1, CR3, CD4 etc for efficient complement-mediated phagocytosis. Unlike findings in this study, phagocytosis of other Mycoplasma species is CR-dependent. Mmm-specific serum mediated phagocytosis and the lack of phagocytosis in the presence of cytochalasin-D is also interesting and informative finding. It would have been ideal if authors used heat-inactivated antiserum to clearly indicate involvement of FC-receptors. Additionally, authors could have also adsorbed Mmm-specific Abs in the antisera and then perform the phagocytosis assays to definitively confirm the role of Ab. Similar studies have been done for M. bovis (Prysliak et al., 2023). The suggestion of C-type lectin receptor mediated entry into macrophages warrant additional studies. As authors have described in line 349, it is intriguing why the MIB-MIP system failed to cleave antibodies. If the whole genome sequencing data is available to Mmm 8740-Rita strain, authors could check for the presence (or absence) of MIB-MIP genes.

6. PLOS authors have the option to publish the peer review history of their article (what does this mean?). If published, this will include your full peer review and any attached files.

Reviewer #1: No

Reviewer #2: **Yes: **Rohana P. Dassanayake

---

## [Author Response · Author response to Decision Letter 0]

19 Feb 2024

Answers to reviewer #1:

1/ Line 108-109: In this manuscript, the methodology used to study the interaction of Mmm and macrophages is extremely important. So, Mmms were left with the macrophages for 1 hr in all the experiments? 

 Yes.

2/Were the cells washed before adding different serums. 

 Yes since the medium was replaced as stated line 107.

3/The way methodology is written currently, it seems that both mycoplasmas and serums were added together. It is very important for the results observed in Fig 2 because if the mycoplasmas and serums were added together then its really not different compared to the data presented in Fig 1. 

The authors need to clarify this. 

 Experiments in Fig1 are performed in the absence of macrophages whereas in Fig 2 mycoplasmas and serums are added to macrophages. In addition, bovine sera were used at non-bactericidal concentrations in Fig2’s experiment. Incubation of mycoplasmas with bovine sera for 30min before addition to macrophages was also tested and gave similar results (see lines 356-358 in first version).

4/ Flow cytometry analysis (Line 130-132): The flow cytometry analyses are important in order to understand whether Mmms are internalised (gone inside of the macrophages) or associated (made complex and stuck outside) with macrophages. Based-on the current gating strategy how would you differentiate macrophage-associated Mmm from macrophage-internalised Mmm?

 By the use of propidium iodide (PI) as stated lines 144-145. See also answer to question 4/ below.

5/ Given that Mmms are so small, it will be very challenging to differentiate the macrophage-associated Mmm from macrophage-internalised Mmm. Due to this limitation, both the externally bound Mmm (large proportion) and internalised (minimum proportion) to macrophages will generate the same double positive signal (FITC + PI). 

 As stated in sub-chapter “flow cytometry” of chapter “Mat&Met”, propidium iodide (PI) is added after harvesting macrophages after 1h incubation with mycoplasmas, and at 4°C, and for 10min only. In other words, and since PI does not enter viable macrophages, the only situation where internalized Mmms would also be positive for PI is if they are taken up by macrophages during that 10min incubation time at 4°C which prevents phagocytosis. In contrast, the majority of mycoplasmas internalized during the 1h incubation at 37°C in the absence of PI cannot be positive for PI.

6/ Line 180: Based on the methodology, it doesn't seem that the authors have focused on Mmm associated to the macrophages. Please clarify. 

 In flow cytometry, a size-based gate is commonly used to exclude cell-debris. Such a gate will also contain mycoplasmas due to their small size. As explained above, larger mycoplasmas aggregates will be positive for PI and excluded from the analysis.

7/Line 181: The authors mention that "no cytopathological effects were detected on macrophages". Did the authors measure viability of the macrophages?

 No, only adherence and morphological features were observed under a light microscope. Since macrophages adhere to and spread on plastic, any cytopathological effect would result in cell rounding and/or loss of adhesion.

8/Line 186: This is bit confusing. MOI is always represented in relation to cells, but the authors are saying in the absence of macrophages. Please clarify. 

 A supplementary figure (S2 Fig) was added to show results obtained in the absence of macrophages.

9/Line 205: How was this conclusion made? The authors just mentioned that that "overwhelming majority of Mmm survival at 24 h was due to extracellular mycoplasmas". I don't think that the authors can make this conclusion based on the current results. First, it needs to be demonstrated that Mmm can survive in macrophages. Given that very low number of Mmm were detected in the G+ treatment group, it is almost impossible to say that "complement had no effect on Mmm survival inside macrophages".

 The gentamicin assay is a classical assay used since many years to address the question of intracellular survival of mycoplasmas. Only intracellular bacteria will survive in the presence of gentamicin in the medium. Here we show that CFU titers in mixed Mmm+macrophages cultures dramatically drops when gentamicin is added to the medium. Thus, indicating that the vast majority of Mmm survival at 24 h was due to extracellular mycoplasmas. If complement had a significant effect on intracellular survival of Mmm, it would substantially increase the CFU titer which is not the case.

10/ Line 220: These results are contradictory to the previous observations presented in Fig 3. In that, hardly any Mms were internalised but here the authors claim that over 90% were engulfed by the macrophages. Please clarify these contradicting results. 

 Fig 3 does not give information on the amount of Mmm being internalized but on Mmm survival inside macrophages. In Fig 4, fluorescence of macrophages due to internalization of neongreenMmm was observed but only at high MOIs (i.e., MOI>1:100, which is 10 times the MOI used in Fig3’s experiment). Unfortunately, intracellular survival could not be assessed at that MOI since gentamicin was not 100% bactericidal. This has been addressed in the discussion lines 334-337 of the old version of the manuscript: “Unfortunately, gentamicin was not 100% bactericidal at MOIs used for flow cytometry, thus preventing assessment of the viability of intracellular Mmm. The lack of decrease in fluorescence over time suggests that either no killing occurred or that it was not sufficient to cope with ongoing growth of Mmm on macrophages.”.

11/ Line 222: While it is understandable the rational of using neogreen expressing strain, I think there is bigger fundamental problem here. Given that no FITC signal was detected when MOI of 100 or less were used, it clearly indicates that both the WT and neogreen expressing strains are not behaving the same when it comes to their ability to infect or engulfed by macrophages. The first question, which must be answered that how comparable these two different strains are in terms of their ability to internalise or engulfed by macrophages. As far as I see, there is no data on this. Therefore, no conclusion can be made from any comparisons of the mutant and WT Mmm strains in this study. 

 The WT Mmm are used to obtained the background level of fluorescence. The lack of fluorescence above that background obtained with neongreen Mmm at an MOI of 100 does not exclude engulfment by macrophages. It can simply be a matter of sensitivity of the assay due to high background fluorescence of WT Mmm. Nevertheless, at higher MOIs, neongreen Mmm gave up to one-log increase in fluorescence which is very significant, albeit low, suggesting low numbers of Mmm being engulfed by each macrophage.

12/Line 226: Given that the uptake (internalisation) is low, this conclusion cannot be made. 

Line 242: Fig 4 By looking at the graphs, background fluorescence levels (hatched line) appear to be changing. Moreover, it should be at "zero" and not 10^2 or 10^3. It seems like that flow cytometry gating wasn't done properly and the gating strategy must be presented as the supplementary material.

 This has nothing to do with gating but is due to the choice of a logarithmic scale for FITC fluorescence. Lowering the sensitivity setting for FL1/FITC was tested and indeed allowed calibrating the peak signal given by WT Mmm around zero. However, this had the effect of compressing the signals and was masking the slight increase in fluorescence produced by neongreen Mmm in the absence of antiserum.

13/ Line 275: Mmm were diluted in PBS. Can the authors provide the source of PBS and other reagents used in the cell-culture experiments? Given that TNF levels can be easily induced even by a small amount of co-contaminating endotoxin levels in reagents, it is absolutely important to use certified endotoxin free reagents in cell culture experiments. 

 PBS alone was tested in pilot experiment and shown to have no effect on background TNF levels in supernatant from macrophages cultures.

14/Line 278-279: I don't think, that is the correct way of reporting experimental findings. That's the variability of the outbred animals. You can't just take two best readings out of three. The results from the third animal must be reported. Otherwise, this experiment becomes, redundant. 

 There was no effect of Mmm infection with or without complement on that one animal most likely because the background control (i.e., medium+FCS only) was very high. Therefore, it was removed from the analysis. It is common when using large outbred animals to exclude those that show activation in the absence of stimuli because due to financial constraint small animal numbers are used.

15/Line 292: Fig 5 so, 2 animals each for serum 1 and serum 2? Is that correct?

 Yes that is correct.

16/Line 315: Yes, this is a most likely scenario, but authors kept saying that Mmms have been internalised. So, results need to be updated. 

 No, because there can be internalization without intracellular survival

17/Line 319: The authors should avoid using "internalisation or intracellular" because based on the results presented these conclusions can't be made. The current flow cytometry analyses are not sufficient to conclude that Mmms are internalised into macrophages. Other methodology such as confocal microscopy could be used to analyse internalisation of Mmm into macrophages. 

As explained above, the flow cytometry method used in our study is tailored to target intracellular Mmm and has been validated previously by our group using confocal microscopy

 (DOI: 10.1016/j.jbiotec.2016.08.006)

18/Line 345: But the authors just mentioned (in line 339) that the uptake of Mmm was independent of phagocytosis. Please clarify and discuss these in the discussion section.

 This section refers to cultures were anti-Mmm antiserum (specific opsonization) has been added whereas line 339 refers to the presence of bovine complement (non-specific opsonization).

19/Line 346: Again, the intensity can also increase if Mmm + macrophages form a complex and not necessarily internalised. So, the gating strategy must be presented with side and forward scatter graphs. 

 See explanation above for exclusion from the analysis of extracellular Mmm bound to macrophages membranes.

20/Line 352: Why did the authors not comment on the third animal, which showed substantial increase in TNF levels. The impact of that on the study should be discussed here. 

 That animal was removed from results due to high background responses (see point 14).

Answers to reviewer #2:

Abstract: 

1/Its better to mention monocyte-derived macrophages (MDMs) in the abstract (line: 14, 21, 22, and rest of the manuscript). Because authors used MDMs rather than macrophages isolated from lung lavage and/or peritoneal cavity for the phagocytosis assay.

 Macrophages has been replaced by MDMs throughout the manuscript. 

2/It would be more informative to mention authors measured TNF-alpha by ELISA (abstract and other places) rather than just typing TNF as it has been mentioned in Tette et al., 2015 publication. 

 TNF has been replaced by TNF-alpha throughout the manuscript.

Materials and Methods: 

1/Line 74: Define PBS. Done.

2/Line 75: change , or CFU/mL to (CFU/mL), Done.

3/Line 76: define PPLO. Done.

4/Line 78: start the sentence with Twenty uL (not 20uL), were the tubes prepared in triplicate and/or spotted three 20uL on the plate? Done. Experiments were performed in triplicates not titration. 

5/Line 83: this author is not familiar with the term “non-decomplemented bovine sera” but rather familiar with “non-heat-inactivated sera”. “Non-heat-inactivated” was added line 93.

6/Line 88: remove at 19-23C. Done.

7/Line 93-94: change “The specific Mmm antiserum” to “The Mmm specific antiserum”. Done.

8/Line 97: this author usually writes “complement was inactivated by heating….” rather than “decomplemented ….”. 

 The sentence was changed accordingly in lines 109-110.

9/Line 99: change Macrophages in the title to “Monocyte-derived macrophages”. Done.

10/Line 109: define MOI and also typically MOI is written as 1:1; 1:10; or 1:100. This author typically write as below “multiplicity of infection (MOI) at a 1:10 ratio (monocytes: bacteria)”. 

 The sentence was changed lines 124-125 and elsewhere in the manuscript.

11/Line 111: did the authors use any other percentage of sera for the assays such as 50% v/v and 90% v/v etc. 

 No since the objective of the study was to use sera at non-bactericidal concentrations.

12/Line 114: why did not authors use Trypsin/EDTA or cold EDTA to lift the cells?

 That would require an additional step to wash the cells before adding Trypsin in order to remove trace amount of FCS that is present in the medium.

13/Line 127: what is the final concentration of PI? (6 ug/mL?)

 100 ug/mL.

15/Line 135: authors could define excitation and emission wave lengths for GFP and PI and the band width of filters used. 

 This information can easily be found on the internet or on the manufacturer’s websites. 

16/Line 146: TNF-alpha? Done.

17/Line 152-155: its better to describe here how the data were presented such as mean and standard deviations (SD) [of CFU or optical densities (OD)] etc.

 This information is already provided in each sub-chapter of the materials and methods chapter.

Results: 

1/Line 165: is direct cytotoxicity of complement on Mmm due to the formation of membrane attack complex through alternative and/or C-type lectin activation complement pathway? 

2/Line 166: since serum heat inactivation condition is described under methods (line 97), 56C, 30 can be removed and modify the sentence accordingly. Move the reference to line 97. 

 Done.

3/Line 176 and in other figures, just indicate * and P value since authors have already defined significance under the statistics section (line 154-155). 

 Done.

4/Line 182-184: in Fig. 1a, authors have clearly demonstrated significant killing activity with 10% serum samples from two of the three animals during 1 hr incubation. It is not clear to this reviewer then why the 10% serum had no effect when in the Fig. 2. Also why did authors used 5-10% serum vs 20% Mmm-specific antiserum knowing that 20% non-decomplemented serum can kill Mmm unless antisera were heat-inactivated (but this reviewer could not find this information from the method).

 Serum from these animals were used at 5% in Fig2’s experiment. Mmm-specific antiserum was heat inactivated prior to use and as mentioned line 97. 

5/Line 189: are you suggesting here killing of Mmm with specific antisera are due to the activation of classical compliment pathway (and/or Fc-receptor mediated phagocytosis)? 

 Fc-receptor mediated phagocytosis since Mmm-specific antiserum is heat-inactivated.

6/Line 217-239: was the Mmm-specific antisera heat inactivated? If so, this phagocytosis most likely through Fc-receptor pathway.

 Yes. 

7/Line 273-282: it’s better to mention authors measured TNF-alpha here because mAb used were TNF-alpha specific (Totte et a., 2015 study) 

 Done.

8/Line 273-282: the lack of TNF-alpha at lower MOI is intriguing since Sacchini et al., 2012 indicated the presence of several cytokines including TNF-alpha when cattle were challenged with Mmm. Does this mean TNF-alpha is produced by other type of cells under lower MOI (such as B-cells, NK cells, endothelial cells, fibroblast etc.?

 It is difficult to know precisely what are the MOIs in vivo in infected alveolar spaces. Mmm concentrations in pleural fluid of diseased animals were reported to reach up to 1012 CFU/mL whereas, in this study, 1.5 x 107 to 1.5x108 CFU/mL were used to observe TNF-alpha production by infected MDMs. It would be very interesting to study interactions of Mmm with other immunocompetent cells.

Discussion: 

1/It would have been ideal if authors used heat-inactivated antiserum to clearly indicate involvement of FC-receptors. 

 This was the case in our study as explained above in points 4-6 of “results”.

2/Additionally, authors could have also adsorbed Mmm-specific Abs in the antisera and then perform the phagocytosis assays to definitively confirm the role of Ab. Similar studies have been done fo

---

## [Decision Letter · Decision Letter 1]

2 Apr 2024

PONE-D-23-32412R1

Interactions between Mycoplasma mycoides subsp. mycoides and bovine macrophages under physiological conditions

PLOS ONE

Dear Dr. Totté,

Thank you for submitting your manuscript to PLOS ONE. After careful consideration, we feel that it has merit but does not fully meet PLOS ONE’s publication criteria as it currently stands. Therefore, we invite you to submit a revised version of the manuscript that addresses the points raised during the review process.

Please submit your revised manuscript by May 17 2024 11:59PM.  If you will need more time than this to complete your revisions, please reply to this message or contact the journal office at plosone@plos.org. Please include the following items when submitting your revised manuscript:

We look forward to receiving your revised manuscript.

Kind regards,

Rohana P Dassanayake

Guest Editor

PLOS ONE

Journal Requirements:

2. We noted in your submission details that a portion of your manuscript may have been presented or published elsewhere. [the article was preprinted and recommended by PCI https://microbiol.peercommunityin.org/articles/rec?id=5] Please clarify whether publication was peer-reviewed and formally published. If this work was previously peer-reviewed and published, in the cover letter please provide the reason that this work does not constitute dual publication and should be included in the current manuscript.

Reviewers' comments:

Reviewer's Responses to Questions

**Comments to the Author**

1. If the authors have adequately addressed your comments raised in a previous round of review and you feel that this manuscript is now acceptable for publication, you may indicate that here to bypass the “Comments to the Author” section, enter your conflict of interest statement in the “Confidential to Editor” section, and submit your "Accept" recommendation.

Reviewer #3:

Reviewer #4: All comments have been addressed

2. Is the manuscript technically sound, and do the data support the conclusions?

Reviewer #3: Yes

Reviewer #4: Yes

3. Has the statistical analysis been performed appropriately and rigorously?

Reviewer #3: Yes

Reviewer #4: Yes

4. Have the authors made all data underlying the findings in their manuscript fully available?

Reviewer #3: Yes

Reviewer #4: Yes

5. Is the manuscript presented in an intelligible fashion and written in standard English?

Reviewer #3: Yes

Reviewer #4: Yes

6. Review Comments to the Author

Reviewer #3: Comments to authors

The article investigated the interactions between Mycoplasma mycoides subsp. mycoides and bovine monocyte-derived macrophages, in presence or absence of different concentrations of complement or anti-Mmm antiserum.

The research provides a relevant contribution to further understanding the mechanisms of host-pathogen interaction that occur in the early stages of contagious bovine pleuropneumonia (CBPP). Macrophages (namely alveolar macrophages) represent the first line of defence in the lung where the infection takes place. The results demonstrated that Mmm specific antibodies enhance phagocytic and bactericidal activity of MDMs. Importantly these data confirm that high concentrations of Mmm are able to directly induce macrophages to produce TNF-alpha, a pro-inflammatory cytokines that correlate with severe CBPP disease.

In general the article is well written and the methodology applied was functional to answer the research questions investigated.

Results are clearly presented.

Minor comments.

Conclusions

Considering the very interesting work done, I would recommend the authors to integrate the discussion commenting the results within the context of CBPP pathogenesis to increase the significance and the importance of the research. Find below some aspect to consider:

Use of MDMs. One of the main challenge in CBPP research is the difficulty of investigating the host-pathogen interaction at lung level in the early stages of disease onset. Due to the lack of laboratory animal models of CBPP infection, we urgently need simplified in vitro and ex vivo models to dissect the complex cattle immune response leading to disease pathology. Even if the ideal target cells to be investigated are the alveolar macrophages, harvesting of these cells from in vivo is not straightforward and require expert animal manipulation. The use of MDMs represent a valid alternative and it provided similar results with the advantage of starting from a simple blood sampling.

Complement. What is the current knowledge on the role of complement in the lung? May the complement contribute to down modulate the macrophage activation during the primary phase of infection and mycoplasma replication? Or the abrogation of TNF-alpha production by MDMs observed in presence of bactericidal concentrations of complement is simply related to the reduction of Mmm concentration?

TNF-alpha. High concentrations of Mmm can induce TNF-alpha release by macrophages even in the absence of anti-Mmm antiserum. May this explain why there is no correlation between severe pathological findings and antibody response in CBPP infected animals? What is the role of these cytokine in cbpp pathogenensis and lung pneumonia? More importantly, what is the immune networking that is activated by tnf-alpha? Can the induction of TNF-alpha initiate the immune mechanisms that lead to CBPP fibrinous pneumonia?

Furthermore only high concentration of Mmm induced production of TNF-alpha. This indicates that while Mmm is replicating at low concentrations the immune system is not activated. May this explain why, in natural conditions, naïve animals requires multiple or prolonged exposure to diseased animal to get infected? Similarly, may this support the long incubation time sometimes observed in infected animals?

Other comments

Minor editing corrections are required in the paragraph results Effect of bovine complement on the uptake of fluorescent Mmm by macrophages: line 232-257 (some errors: of of, MIF, FIg4c)

Reviewer #4: Comments to authors:

Based on my review, I find the manuscript acceptable for publication. The authors have responded to the previous reviewer's comments well.

The knowledge generated from this study is crucial in enhancing our understanding of the Mmm interaction with macrophages.

Abstract:

Line 202: Write the abbreviation “MOI” in full

Introduction:

Line 36: Write WOAH in capital letters

Results:

Line 181: the sentence is not clear. It may be replaced with … “The mycoplasmacidal activity and inter-animal variability of bovine sera remained consistent throughout the study”.

Line 215: The efficacy of antibiotics against mycoplasmas is restricted to specific mycoplasma species and only reduces the concentration of mycoplasmas rather than killing the bacteria. Why was gentamycin used when there are more effective antibiotics like Tyrosin?

Discussion:

Line 336: It’s not clear. This information indicates that gentamicin was effective in preventing the survival of the bacteria within the macrophages, leading to the conclusion that the majority of the viable bacteria were located on the cell surfaces. Why was the gentamycin effective in this case?

Line 376: Use a uniform method of citation.

7. PLOS authors have the option to publish the peer review history of their article (what does this mean?). If published, this will include your full peer review and any attached files.

**Do you want your identity to be public for this peer review? **For information about this choice, including consent withdrawal, please see our Privacy Policy.

Reviewer #3: No

Reviewer #4: Yes 

Do you want to get recognition for this review on a Web of Science researcher profile?

If you opt in, your Web of Science profile will automatically be updated to show a verified record of this review in full compliance with the journal’s review policy. If you don’t have a Web of Science profile, you will be prompted to create a free account.

Reviewer #3: Yes

Reviewer #4: Yes

---

## [Editor Report · Decision Letter 2]

6 Jun 2024

Interactions between Mycoplasma mycoides subsp. mycoides and bovine macrophages under physiological conditions

PONE-D-23-32412R2

Dear Dr. Totte,

We’re pleased to inform you that your manuscript has been judged scientifically suitable for publication and will be formally accepted for publication once it meets all outstanding technical requirements.

Kind regards,

Rohana P Dassanayake

Guest Editor

PLOS ONE
---

## [Editor Report · Acceptance letter]

18 Jun 2024

PONE-D-23-32412R2 

PLOS ONE

Dear Dr. Totté, 

I'm pleased to inform you that your manuscript has been deemed suitable for publication in PLOS ONE. Congratulations! Your manuscript is now being handed over to our production team.

Kind regards, 

on behalf of

Dr. Rohana P Dassanayake 

Guest Editor

PLOS ONE